# Visualization of Germination Proteins in Putative *Bacillus cereus* Germinosomes

**DOI:** 10.3390/ijms21155198

**Published:** 2020-07-22

**Authors:** Yan Wang, Richard de Boer, Norbert Vischer, Pauline van Haastrecht, Peter Setlow, Stanley Brul

**Affiliations:** 1Molecular Biology and Microbial Food Safety, Swammerdam Institute for Life Sciences, University of Amsterdam, Amsterdam, Science Park 904, 1098 XH, The Netherlands; y.wang5@uva.nl (Y.W.); R.deBoer@uva.nl (R.d.B.); norbertvischer@gmail.com (N.V.); PaulinevanHaastrecht@outlook.com (P.v.H.); 2Department of Molecular Biology and Biophysics, UConn Health, Farmington, CT 06030-3305, USA; setlow@uchc.edu

**Keywords:** *Bacillus cereus*, spore, germinant receptor, germination protein, spore inner membrane

## Abstract

*Bacillus cereus* can survive in the form of spores for prolonged periods posing a serious problem for the manufacture of safe shelf-stable foods of optimal quality. Our study aims at increasing knowledge of *B. cereus* spores focusing primarily on germination mechanisms to develop novel milder food preservation strategies. Major features of *B. cereus* spores are a core with the genetic material encased by multiple protective layers, an important one being the spores′ inner membrane (IM), the location of many important germination proteins. To study mechanisms involved in germination of *B. cereus* spores, we have examined the organization of germinant receptors (GRs) in spores′ IM. Previous studies have indicated that in spores of *B.*
*cereus* ATCC 14579 the L-alanine responsive GR, GerR, plays a major role in the germination process. In our study, the location of the GerR GR subunit, GerRB, in spores was examined as a C-terminal SGFP2 fusion protein expressed under the control of the *gerR* operon′s promoter. Our results showed that: (i) the fluorescence maxima and integrated intensity in spores with plasmid-borne expression of GerRB-SGFP2 were significantly higher than in wild-type spores; (ii) western blot analysis confirmed the expression of the GerRB-SGFP2 fusion protein in spores; and (iii) fluorescence microscopy visualized GerRB-SGFP2 specific bright foci in ~30% of individual dormant spores if only GerRB-SGFP2 was expressed, but, noticeably, in ~85% of spores upon co-expression with GerRA and GerRC. Our data corroborates the notion that co-expression of GR subunits improves their stability. Finally, all spores displayed bright fluorescent foci upon expression of GerD-mScarlet-I under the control of the *gerD* promoter. We termed all fluorescent foci observed germinosomes, the term used for the IM foci of GRs in *Bacillus subtilis* spores. Our data are the first evidence for the existence of germinosomes in *B. cereus* spores.

## 1. Introduction

In general, bacterial endospore formers comprise aerobically growing Bacillales species and anaerobic Clostridiales species. Bacillales species are ubiquitous in nature and can be found in soil, plants, insects and mammals. As such, they can easily contaminate food products and are often causative agents of foodborne disease [1]. For example, *Bacillus cereus* can lead to an emetic or diarrheal syndrome as a result of human consumption of contaminated foods in which outgrowing spores and their resulting vegetative cells have formed toxins [2]. Spores can survive for years or even decades in harsh conditions, including wet or dry heat, high pressure and radiation dense areas, due to spores′ novel structure and many protective components and their metabolic dormancy [3,4].

In *B. subtilis*, a generally used model spore-former, the initiation of sporulation is largely dependent upon transcription factor Spo0A [5]. After this initiation, sporulating cells divide into a larger mother cell and a smaller forespore, and then the mother cell engulfs the forespore [6,7,8]. After engulfment, the forespore matures and ultimately is released from the mother cell as a highly resistant free spore in which metabolic activity has been silenced. The structure of a mature *B. cereus* spore is characterized by an outermost exosporium, then the outer and inner coat layers, an outer membrane, the large cortex peptidoglycan layer, then a layer of germ cell wall peptidoglycan and finally the inner membrane (IM). The IM contains typical plasma membrane phospholipids as well as a number of proteins, some of which are involved in spore germination, and lipid mobility in the dormant spores′ IM is extremely low [9]. The core contains the DNA, essential macromolecules for survival and protective calcium dipicolinic acid (CaDPA).

Studies in *B. subtilis* identified the proteins that are most likely to interact somehow with physiological spore germinants, and these proteins have been termed germinant receptors (GRs). There are three major *B. subtilis* GRs-GerA, GerB and GerK-, and all are present in the IM and are made-up of A, B and C subunits [10,11]. Generally, subunit A is comprised of five or six predicted transmembrane (TM) domains and hydrophilic domains at the N-and C-termini, but the B subunit is comprised of only 10–12 TM domains. The A and B GR subunits are the ones most likely to be involved in recognition of amino acid germinants [12]. The GR C-subunit includes a predicted pre-lipoprotein signal sequence, suggesting that the C subunit is anchored to the outer surface of the membrane via an N-terminally attached lipid moiety [13]. Spores are triggered to germinate in response to specific amino acids, sugars and purines. Upon GR triggering by a germinant the germination process proceeds beyond a point of no return (commitment) and leads to the release of CaDPA, hydrolysis of cortex peptidoglycan by cortex-lytic enzymes, the swelling of the core and finally full core hydration [10]. After this change, germinated spores enter into outgrowth that leads to the first vegetative cell division.

*Bacillus cereus* is a member of the Gram-positive, motile, rod-shaped endospore forming *Bacillus cereus* group; other well-known members of the *B. cereus* group are *B. thuringiensis* [14], *B. anthracis* [15], *B. mycoides* [16], *B. weihenstephanensis* [17], *B. pseudomycoides* [18], *B. toyonensis* [19] and *B. cytotoxicus* [20]. In *B. subtilis* spores, three types of proteins related to germination are located in the IM: GRs, GerD acting as a scaffold protein is almost certainly a lipoprotein [21] and SpoVA channel proteins for CaDPA. There are three GRs: GerA responding to L-alanine, and GerB and GerK that collaborate in responding to L-asparagine, D-glucose, D-fructose and potassium chloride [10]. In contrast, there are seven functional GRs—GerR, GerK, GerG, GerL, GerQ, GerI and GerS—in *B. cereus* ATCC 14579 spores [22]. In particular, GerR, with the order GerRA-GerRC-GerRB, has a critical role in spore germination in response to L-alanine, or inosine alone and in food products, for example meat broth and rice water [23].

In *B. subtilis* spores, GRs are localized in 1–3 IM foci, termed a germinosome, for which GerD acts as a scaffold for GR assembly [21,24]. Recent work by our laboratory, using super-resolution three-dimensional structured illumination microscopy (3D-SIM) as well as annular Rescan Confocal Microscopy has allowed us to quantify germinosomes in single spores as well as to monitor germinosome dynamics upon germinant addition to wild-type *B.* subtilis spores [25,26].

In contrast to identification of germinosomes in *B. subtilis* spores, nothing is known about the location of germination proteins in *B. cereus* spores. Hence, here we aimed to use phase-contrast microscopy and the Nikon Eclipse Ti fluorescence microscope to see if we could visualize putative germinosomes in *B. cereus* spores, initially using a strongly enhanced green fluorescent protein (SGFP2) fused to the GerRB germination protein, and under the control of the native promoter of the *gerR* operon on a low-copy number plasmid to enhance chances of observing the fusion protein in wild-type spores. The results with this fusion were compared to those with spores of a *B. cereus* strain in which the plasmid contained the *gerRB-SGFP2* fusion gene without the *gerR* promoter. Initial results with this first GerRB-SGFP2 fusion were promising, as ~30% of spores carried likely germinosomes, but this result suggested using a plasmid carrying the entire *gerR* operon with *gerRB* fused to *SGFP2* and under P*gerR* control. This plasmid gave much higher levels of GerRB-SGFP in spores, and the protein was clearly in one or multiple foci in ~85% spores. In addition, a plasmid carrying *gerD* plus its promoter fused to a gene encoding the mScarlet-I red fluorescent protein also gave spores with red foci in all spores, consistent with this protein also being localized in germinosomes in *B. cereus* spores.

## 2. Results

### 2.1. Visualization of Germinant Proteins in a Germinosome in B. cereus Dormant Spores

*B. cereus* strains were constructed which expressed either the GerRB protein or all GerR proteins fused with strongly enhanced green fluorescent protein (SGFP2) at the GerRB C-terminus (Table 1). Note that a study of the topology of a *B. anthracis* GR B subunit found that this protein′s C-terminus was inside the spore′s IM [27]. A series of pHT315 recombinant plasmids and pHT315-derived *B. cereus* strains were constructed and their proper construction was confirmed using restriction enzyme digestion and colony PCR (Figure 1; Appendix A). The strategy involved cloning the promoter sequence of the *gerR* operon (Appendix A) upstream of the entire *gerRB* coding sequence without its stop codon and followed by a small-region coding for a linker and then the *SGFP2* gene. The high level GerRB-fusion protein expression would allow competition for presumed binding sites in the spore for native GerRB protein expressed from the chromosome. As controls, strains carrying plasmids either with expression of SGFP2 protein alone controlled by the constitutive promoter of the *aphA3′* (aminoglycoside phosphotransferase A3′) gene or with the empty vector pHT315 were constructed, as was a strain with a *gerRB-SGFP2* but no promoter. An additional pHT315 plasmid was constructed in which the entire *gerR* operon was under P*gerR* control, with *gerRB* fused to *SGFP2*. With this construct, the polycistronic *gerR* mRNA would promote translational coupling between *gerRA*, *gerRC* and *gerRB-SGFP2*, and thus GerRB-SGFP2 is more likely to be incorporated into a GR than if it were translated alone. Indeed, GR subunits translated alone have been found to be rather unstable [28]. One final plasmid was constructed in which *gerD* and its promoter were fused to the *mScarlet-I* gene; note that based on results in *B. subtilis*, the *gerD* promoter is likely stronger than the *gerR* operon′s promoter [29].

To examine the expression of the fusion proteins in spores, freshly sporulated *B. cereus* spores were purified and analyzed by fluorescence microscopy. The maximum and integrated fluorescence intensity of 228 individual spores was measured using ImageJ software. The results indicated that the maximum and integrated intensities from spores carrying the GerRB-SGFP2 and GerR-SGFP2 fusion proteins expressed from genes under *gerR* control were both significantly higher than values for the wild-type spores and spores carrying pHT315. Note that there seems less general diffuse fluorescence if the entire operon is expressed (compare Figure 2H,J) corroborating better germinosome localization of GerRB-SGFP2 upon co-expression. Results obtained for spores carrying the GerD-mScarlet-I fusion protein under *gerD* control were in line with the GerRB-SGFP2 data though the areas of high fluorescence in the spores vary somewhat morphologically (Figure 3). The spores carrying pHT315 also showed significantly lower fluorescence maximum intensity than wild-type spores (Figure 3A), but there was no significant difference in these spores′ integrated fluorescence intensity (Figure 3B). However, the maximum and integrated fluorescence intensity of spores carrying pHT315-P*gerR*-*gerRB*-*SGFP2* were significantly higher than in spores carrying pHT315-*gerRB*-*SGFP2* with no *gerR* promoter (Figure 3). We then quantified the number of bright spots or foci, which we have designated as germinosomes (Figure 4). As is shown, such structures were visualized in some individual spores obtained from cells carrying plasmids with *gerRB-SGFP2*. When a total of 2075 spores were analyzed in three separate experiments (516, 792 and 767 spores), this analysis showed that the germinosome was present in 32% of spores carrying pHT315-P*gerR*-*gerRB*-*SGFP2* (Figure 4). A total of 915 spores carrying the entire *gerR* operon with *gerRB* fused to *SGFP2* were analyzed; this analysis showed that one and two germinosomes were present in 71% and 14% of the spores carrying pHT315-P*gerR*-*gerR*-*SGFP2*, respectively (Figure 4) A total of 750 spores carrying pHT315-P*gerD*-*gerD*-*mScarlet-I* were also analyzed; this analysis showed that one, two and three germinosomes were present in 36%, 37% and 27%, respectively, of the spores (Figure 2, Figure 4 and Figure 5E). Notably, in *B. subtilis* spores, initial visualization of the germinosome was hampered by high autofluorescence from coat proteins [29,30,31], but *B. cereus* spore coat protein autofluorescence seems much lower.

### 2.2. Validation that All Spores Carry Plasmids

One concern in using a plasmid-encoded fusion protein gene to visualize potential germinosomes in spores, is that plasmid partition at the sporulation division may or may not be 100% [32,33,34]. Consequently, many spores from plasmid-carrying cells may not carry the gene for the fusion protein. To examine this possibility, 24 colonies, each generated by single spores obtained from *B. cereus* cells carrying pHT315-P*gerR*-*gerRB*-*SGFP2*, were examined on TSB-agar plates containing 10 μg/mL erythromycin, and all grew. In addition, the expected DNA fragment of P*gerR*-*gerRB*-*SGFP2* was amplified successfully from 24 spore-derived colonies of *B. cereus* strain 003 carrying pHT315-P*gerR*-*gerRB*-*SGFP2* (Figure 6). These results indicate that all dormant spores obtained from *B. cereus* cells carrying pHT315-P*gerR*-*gerRB*-*SGFP2* retained at least one copy of the plasmid.

### 2.3. Confirmation of Fusion Protein GerRB-SGFP2 Expression in B. cereus Dormant Spores

We designed pHT315-derived *B. cereus* strains to visualize the GerR B-subunit fused to a GFP-reporter and under the control of the promoter of the *gerR* operon. The fusion protein contained a flexible linker between the *gerRB* and *SGFP2* genes and lacked the *gerRB* stop codon. western blotting with anti-GFP antibody detected the GerRB-SGFP2 fusion protein at the expected size of 69 kDa in extracts from spores carrying pHT315-P*gerR*-*gerRB*-*SGFP2,* but not SGFP2 alone, while bands corresponding to the size of GerRB-SGFP2 protein were not detected in extracts from wild-type spores or spores carrying pHT315 or pHT315-*gerRB*-*SGFP2* (Figure 7). This indicated that the GerRB protein fused to the SGFP2 protein was expressed under the control of the *gerR* operon promoter. A nonspecific 100 kDa protein in extracts of all spores also cross-reacted with the polyclonal antibodies used, as did a protein of ~55 kDa.

## 3. Discussion

*Bacillus cereus* spores consist of a core with the genetic material surrounded by the protective layers including the IM, a peptidoglycan containing cortex, the proteinaceous coat and the exosporium. Together they provide the spore with extreme resistance. In addition, activation of GRs in the spore′s IM is likely the first event in germination induced by physiological germinants.

Single or double crossovers and overexpression can all be used to fuse a target gene in a bacterial chromosome to the *gfp* reporter gene. Ideally, a double crossover would be used to do this. However, we found that a recombinant plasmid (Appendix A) that would theoretically integrate *gerRB-SGFP2* at the chromosomal *gerRB* gene did not integrate into the *B. cereus* chromosome. Therefore, we generated the fusion gene *gerRB-SGFP2* on the low copy number episomal plasmid pHT315, where expression could be driven by the native promoter of the *gerR* operon. This vector was previously reported to localize the *gerP* operon to the inner coat of *B. cereus* spores and the protein IlsA (iron-regulated leucine rich surface protein) to the bacterial surface of *B. cereus* [35,36]. While this may cause an imbalance in levels of GR subunits which normally are in a 1:1:1 ratio, the higher amount of the GerRB-SGFP2 fusion protein may help GerRB-SGFP2 compete more effectively with wild-type GerRB for association with GerRA and GerRC, as intact GR formation in *B. subtilis* requires all 3 subunits.

Previous studies have shown that the *B. subtilis gerA* operon and the *B. cereus gerR* operon are expressed in developing forespores, and the products are involved in spore germination with L-alanine [23,24]. In our study, the primary objective was to ascertain whether the GerRB-SGFP2 fusion protein can be visualized and localized in *B. cereus* dormant spores, and our work showed that what is almost certainly a germinosome can be visualized by its GerRB-SGFP2 fluorescence in almost a third of *B. cereus* spores (Figure 2H). However, the ~30% of *B. cereus* spores exhibiting a germinosome was much lower than that for germinosomes in *B. subtilis* spores [25,26,29]. Reasons for this large difference in apparent levels of germinosomes are not clear, but there are a number of possibilities including the following. (1) While all spores from cells carrying pHT315-P*gerR*-*gerRB*-*SGFP2* contained at least one plasmid copy, perhaps the actual copy number was very heterogeneous such that ~70% of spores had so few plasmid copies such that insufficient GerRB-SGFP2 was made to compete effectively with wild-type GerRB in GR and germinosome formation. (2) It is known that GR levels in individual spores of at least *B. subtilis* are quite heterogeneous, most likely for stochastic reasons [29,37], and perhaps plasmid-driven GerRB-SGFP2 levels can only compete effectively for GR and germinosome incorporation in spores with low wild type GerRB levels. In terms of possibilities 1 and 2, it is notable that expression of SGFP2 alone from the *gerR* promoter gave very heterogeneous levels of GFP fluorescence in individual spores (Appendix A), although the precise causes of this heterogeneous expression are not clear. (3) When the enteric bacterium *Escherichia coli* is used to express recombinant or heterologous protein, the protein synthesized is sometimes aggregated perhaps because high levels of the recombinant protein synthesized interfere with protein folding [38]. This may be a particular problem for GerRB which is most likely normally co-translated with the other two GR subunits and these may all fold together. However, this may not take place efficiently with plasmid-encoded GerRB, which may be very unstable in the absence of the other GerR subunits. This could greatly decrease the efficiency of GerRB-SGFP2 incorporation into a germinosome in some developing spores. Indeed, the much greater levels of GerRB-SGFP2 and fluorescent germinosomes in spores carrying the whole *gerR* operon with *gerRB*-fused to *SGFP2* strongly suggests that GerRB-SGFP2 instability when expressed alone is most likely a major reason for the low levels of germinosomes in spores only expressing this one GerR subunit. However, the obvious heterogeneity in levels of fluorescence intensities with GerRB-SGFP2, GerR-SGFP2 or GerD-mScarlet-I fusions suggests that heterogeneity in levels of plasmids or GR expression may contribute to this heterogeneity as well.

In conclusion we found that we expressed the GerRB-SGFP2 and GerR-SGFP2 fusion protein driven by the *gerR* operon promoter and the GerD-mScarlet-I fusion protein driven by the gene *gerD* promoter with plasmid to visualize germinosomes and indeed confirmed the existence of germinosomes in *B. cereus* spores for the first time. Our findings lay the groundwork for further research on visualizing the behavior of GR in *B. cereus* spores.

## 4. Materials and Methods

### 4.1. Growth of Bacterial Strains

The strains used in this study are listed in Table 1. *E. coli* DH5α was grown in Lysogeny broth (LB) medium at 37 °C. *B. cereus* strains were grown at 30 °C in trypticase soy broth (TSB) medium. When needed, 100-μg/mL ampicillin or 10-μg/mL erythromycin (Sigma-Aldrich Chemie B. V. The Netherlands) was used.

### 4.2. Isolation of B. cereus ATCC 14579 Genomic DNA

Genomic DNA of *B. cereus* ATCC 14579 was isolated using a modified method from Henderson [39]. Briefly, a single colony was incubated overnight in TSB medium at 30 °C and 200 rpm, centrifuged for 5 min at 12,000 rpm, suspended in the resuspension solution with RNase A (ThermoFisher Scientific, The Netherlands), lysozyme added to 0.2 mg/mL (Sigma-Aldrich Chemie B. V. The Netherlands), and the mix was incubated for 1 h at 37 °C. The suspension was then heated for 5 min at 60 °C, pre-warmed (60 °C) sodium dodecyl sulfate (SDS) solution was added to 1% (*w/v*) (Sigma Aldrich Chemie B. V. The Netherlands), and the lysate was extracted twice by phenol:chloroform:isoamyl alcohol (25:24:1, pH 6.6) (Fisher Scientific B. V. The Netherlands). The upper phase was carefully transferred to a new 2 mL tube, three volumes of absolute ethanol was added and mixed by inverting. Genomic DNA was collected, washed 5 min with 70% (*v/v*) ethanol, dried 10 min at room temperature, dissolved in sterile Milli-Q water and used as a PCR template.

### 4.3. Construction of Recombinant Plasmids

The recombinant plasmids used in this study are listed in Table 2. All primers used are listed in Appendix A. The 676 bp long *gerRB* promoter region (P*gerR*) upstream of the *gerR* operon was PCR amplified from *B. cereus* ATCC 14579 genomic DNA using a pair of primers P_RB-F/P_RB-R. The nucleotide sequence of the *gerRB* (BC_0782) gene was PCR amplified from *B. cereus* ATCC 14579 genomic DNA using primers RB-F/RB-fuse-GFP-R introducing a flexible linker (GSGSGS) [40]. The fragment of *SGFP2* was PCR amplified from plasmid pSGFP2-C1 using primers RB-fuse-GFP-F/GFP-R containing a flexible linker encoding the protein sequence GSGSGS. The purified DNA fragment of P*gerR* was sub-cloned into *Eco*R I and *Bam*H I sites of the shuttle vector pHT315 (Figure 1A), resulting in the pHT315-P*gerR* vector. The *SGFP2* gene was PCR amplified from plasmid pSGFP2-C1 using primers GFP-F/GFP-R, the purified DNA fragment of the *SGFP2* gene sub-cloned into *Xba* I and *Hin*d III sites of pHT315-P*gerR*, resulting in pHT315-P*gerR*-*SGFP2* (Figure 1B). The purified DNA fragments of *gerRB* and *SGFP2* were fused by two-step fusion PCR, the fusion fragment *gerRB-SGFP2* was sub-cloned into *Bam*H I and *Hin*d III sites of pHT315-P*gerR* vector, resulting in pHT315-P*gerR*-*gerRB*-*SGFP2* (Figure 1D). The negative control plasmid pHT315-*gerRB*-*SGFP2* was constructed by sub-cloning the fusion fragment *gerRB*-*SGFP2* into *Bam*H I and *Hin*d III sites of the pHT315 vector (Figure 1F). As a positive control plasmid, the *SGFP2* gene alone was PCR amplified from plasmid pSGFP2-C1 using primers GFP-F/GFP-R, and then the *gfpmut1* gene of vector pHT315-P*aphA3′*-*gfp* [35] was replaced by the purified DNA fragment of the *SGFP2* gene between *Xba* I and *Hin*d III sites, resulting in pHT315-P*aphA3′*-*SGFP2* (Figure 1C). The nucleotide sequence of the *gerRA* (BC_0784) and *gerRC* (BC_0783) genes was PCR amplified from *B. cereus* ATCC 14579 genomic DNA using primers RA-F/AC-fuse-B-R. The *gerRB-SGFP2* fusion fragment was PCR amplified from plasmid pHT315-*gerRB*-*SGFP2* using primers AC-fuse-B-F/GFP-R. Then, these two purified DNA fragments were fused by two-step fusion PCR, the fusion fragment *gerR-SGFP2* was sub-cloned into *Bam*H I and *Hin*d III sites of the pHT315-P*gerR* vector, resulting in pHT315-P*gerR*-*gerR*-*SGFP2* (Figure 1E). The nucleotide sequence of the 606 bp long *gerD* gene promoter (P*gerD*) plus the *gerD* coding gene (BC_0169) was PCR amplified from *B. cereus* ATCC 14579 genomic DNA using primers P_D_F/D-fuse-mSi-R. The *mScarlet-I* gene was PCR amplified from plasmid pEB2-mScarlet-I using primers D-fuse-mSi-F/mSi-R. Then these two purified DNA fragments were fused by two-step fusion PCR, the fusion fragment P*gerD*-*gerD*-*mScarlet-I* was sub-cloned into *Xba* I and *Eco*R I sites of the pHT315 vector, resulting in pHT315-P*gerD*-*gerD*-*mScarlet-I* (Figure 5A). The correct construction of all recombinant plasmids was confirmed using double restriction enzyme digestion and DNA sequencing (Macrogen Europe B. V. The Netherlands).

### 4.4. Preparation of Electro-Competent Cells

*B. cereus* ATCC 14579 cells were cultured overnight in 5 mL TSB medium at 37 °C and 200 rpm. One milliliter of the overnight culture was re-inoculated in 100 mL fresh TSB medium and incubated at 37 °C and 200 rpm until reaching an optical density at 600 nm (OD600) of 0.2–0.3. Cells were washed five times using ice cold electroporation buffer (250-mM sucrose, 1-mM HEPES, 1-mM MgCl_2_ and 10% glycerol, pH 7.0), concentrated 150-fold and 50 μL aliquots were stored at −80 °C until use.

### 4.5. Electroporation

Plasmids pHT315-P*gerR*-*SGFP2*, pHT315-Pg*erR*-*gerRB*-*SGFP2*, pHT315-*gerRB*-*SGFP2*, pHT315-P*aphA3′*-*SGFP2*, pHT315-P*gerR*-*gerR*-*SGFP2*, pHT315-P*gerD*-*gerD*-*mScarlet-I* and pHT315 (Table 2) were introduced into wild-type *B. cereus* competent cells via electroporation, which is performed based on Nathalie′s method [41]. Briefly, 50 μL of *B. cereus* competent cells were thawed on ice, 500 ng plasmid DNA was added, and the mixture was incubated on ice for 5 min. The mixture was transferred into ice-cold 1 mm gap PulseStar electroporation cuvettes (Westburg, The Netherlands, Cat No.: WB 1-4110), and electroporation was performed at 2 kV and 4 ms using a MicroPulser^TM^ electroporation apparatus (Bio-Rad, The Netherlands). After electroporation, 1 mL TSB medium was immediately added to the cells and incubated at 37 °C 200 rpm for 1.5 h. The transformation mixture was plated on tryptic soy agar with 10 μg/mL erythromycin and incubated overnight at 30 °C. The following day an erythromycin-positive single colony was analyzed by colony PCR using sequencing primers 315-F/315-R.

### 4.6. Preparation of Dormant Spores

Sporulation of *B. cereus* strains used in this study was carried out a chemically defined growth and sporulation (CDGS) medium [42], which contained the following components (final concentrations): D-glucose (10 mM), L-glutamic acid (20 mM), L-leucine (6 mM), L-valine (2.6 mM), L-threonine (1.4 mM), L-methionine (470 μM), L-histidine (320 μM), DL-lactate sodium (5 mM), acetic acid (1 mM), FeCl_3_·6H_2_O (50 μM), CuCl_2_·2H_2_O (2.5 μM), ZnCl_2_ (12.5 μM), MnSO_4_·H_2_O (66 μM), MgCl_2_·6H_2_O (1 mM), (NH_4_)_2_SO_4_ (5 mM), Na_2_MoO_4_·2H_2_O (2.5 μM), CoCl_2_·6H_2_O (2.5 μM) and Ca(NO_3_)_2_·4H_2_O (1 mM). The CDGS medium was buffered to pH 7.2 with 100 mM potassium phosphate buffer. As previous described [43], a single colony of *B. cereus* was incubated in 5 mL TSB medium (TSB medium for pHT315-derived *B. cereus* strains had 10 μg/mL erythromycin) at 30 °C and 200 rpm overnight. Then, 200 μL of the overnight culture was transferred into 20 mL fresh TSB medium (TSB medium for pHT315-derived *B. cereus* strains had 10 μg/mL erythromycin) and grown at 30 °C and 200 rpm until the OD600 was 0.5. The cells were harvested by centrifugation at 5000 rpm for 15 min and suspended in 250 mL CDGS medium (CDGS medium for pHT315-derived *B. cereus* strains had 10 μg/mL erythromycin). Finally, the culture was incubated at 30 °C and 200 rpm for 5 d. Spores were harvested, washed three times with pre-cooled phosphate buffered saline (PBS, pH 7.4) (Thermofisher Scientific, Eindhoven, The Netherlands). After harvesting, the spores were further purified by centrifugation through HistodenZ (Sigma Aldrich Chemie B.V., Zwijndrecht, The Netherlands) as follows. One milliliter of spores (OD600 = 100) were transferred to a new 1.5 mL centrifuge tube. After centrifuging, the spore pellet was suspended in 750 μL of 16% HistodenZ which was then slowly layered on 800 μL of 40% HistodenZ and centrifuged for 2 h at 15,000 rpm at 4 °C. After discarding the supernatant, the pelleted spores were suspended in 200 μL of 16% HistodenZ which was slowly layered on 1 mL of 40% HistodenZ and centrifuged for 20 min at 15,000 rpm at 4 °C. The pelleted spores were washed three times with 1–1.5 mL PBS, pH 7.4, to remove HistodenZ. The purified spores contained more than 95% phase bright spores under phase contrast microscopy and were stored at 4 °C until use.

### 4.7. Image Acquisition and Analysis

Preparation of microscope slides and imaging were carried out as described by Pandey [44]. Imaging of dormant spores was performed with a Nikon Eclipse Ti microscope equipped with phase-contrast and fluorescence components. For phase-contrast imaging, a CFI Plan Apochromat Lambda 100X Oil, ORCA-Flash 4.0 Digital CMOS camera C11440–22CU (Hamamatsu Photonics K.K, Sewickley, PA, USA), Prior Brightfield LED and Lambda 10-B shutter (Sutter Instrument, Novato, CA, USA) was equipped with NIS-elements 4.50. All microscope images were analyzed with ImageJ (http://rsbweb.nih.gov/ij), and all statistical analysis were carried out in GraphPad 6.0.

### 4.8. Preparation of Protein Extracts

Extracts of dormant spores were prepared by a published method [45,46]. Spores (OD600 = 50) were suspended in 200 μL TEP buffer (50 mM Tris-HCl pH 7.4, 5 mM EDTA-dipotassium, cOmplete™ Protease Inhibitor Cocktail (Sigma Aldrich Chemie B. V. The Netherlands), 1% Triton X−100). Extracts were generated by bead beating with six rounds of 30 s at 6000 rpm with cooling on ice between each round, in a Precellys 24 tissue homogenizer (Bertin Instruments, Montigny-le-Bretonneux, France) with 800 mg 0.1 mm diameter Zirconia/Silica beads (BioSpec Products, Bartlesville, OK, USA). Lysate samples were incubated with 1% SDS and 150 mM β-mercaptoethanol (Bio-Rad Laboratories B.V., The Netherlands. Cat No.:1610710) for 2 h at room temperature. Samples were bead-beaten twice more for 30 sec at 6,000 rpm to further disrupt aggregates, and 100 μL more TEP buffer was added to further suspend the extracts. Finally, samples were centrifuged for 10 min at 15,000 rpm and 4 °C, and the supernatant fluid was saved. Protein concentrations in extracts were estimated by the Pierce™ BCA Protein Assay Kit (Thermofisher Scientific, The Netherlands, Cat. No.: 23227).

### 4.9. Examination of B. cereus Strain 003 Spores for Plasmid

Dormant spores carrying pHT315-P*gerR*-*gerRB*-*SGFP2* at OD600 of 2.0 were heated 15 min at 70 °C, in duplicate. The heated spores were diluted 10^−5^-fold in TSB medium, 100 μL of diluted spores immediately spread on a TSB agar plate (TSB-agar), and the plates were incubated for 24 h at 30 °C. Following this incubation, 24 single colonies in each replicate were incubated on TSB-agar with 10-μg/mL erythromycin at 30 °C overnight to assess erythromycin resistance plasmid presence was confirmed by colony PCR with the primers 315-F/R.

### 4.10. Western Blotting

30 μg of protein from each extract was mixed with 4× Lemmli SDS sample buffer (Alfa Aesar, The Netherlands, Cat. No.: J60015) and incubated at 65 °C for 30 min. The samples and PageRuler™ Plus Prestained Protein Ladder (Thermo Fisher Scientific, The Netherlands, Cat. No.: 26619) were loaded on a 10% MiniPROTEAN^®^ TGX™ Gel (Bio-Rad, Laboratories B. V., The Netherlands, Cat. No.: 4561034) and separated for 1 h at 180V. Proteins were transferred to a 0.45-μm PVDF membrane by wet electro-blotting for 40 min at 100 V (Bio-Rad, Laboratories B.V., The Netherlands.). The PVDF membrane was blocked with 5% skim milk for 1 h at room temperature and incubated with the rabbit polyclonal anti-GFP antibody at a 1:2500 dilution (abcam, ab290) in PBST (PBS with 0.1% Tween-20) with 1% BSA (Bovine Serum Albumin, Sigma-Aldrich Chemie B. V. The Netherlands) overnight at 4 °C. The membrane was washed with TBST (Tris-buffered saline with 0.1% Tween-20, pH 7.4) five times for 10 min each and incubated with HRP-conjugated goat anti-rabbit IgG H&L at a 1:5000 dilution (abcam, ab205718) in PBST with 1% BSA for 1 h at room temperature, and the membrane was washed with TBST five times for 10 min each. The membrane was then incubated with EZ-ECL substrate solutions (Biological Industries, Cromwell, CT, USA, Cat. No.: 20–500–120) for 2 min at room temperature and detected for 10 min using an Odyssey Fc Imaging system (LICOR, Lincoln, NE, USA).

## Figures and Tables

**Figure 1 ijms-21-05198-f001:**
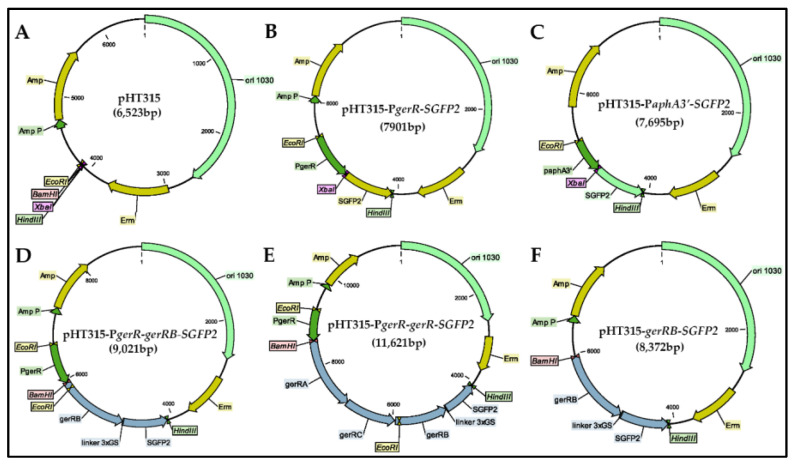
Schematic diagram of the recombinant plasmids. (**A**) pHT315; (**B**) pHT315-P*gerR*-*SGFP2*; (**C**) pHT315-P*aphA3′*-*SGFP2*; (**D**) pHT315-P*gerR*-*gerRB*-*SGFP2*; (**E**) pHT315-P*gerR*-*gerR*-SGFP2; (**F**) pHT315-*gerRB*-*SGFP2*.

**Figure 2 ijms-21-05198-f002:**
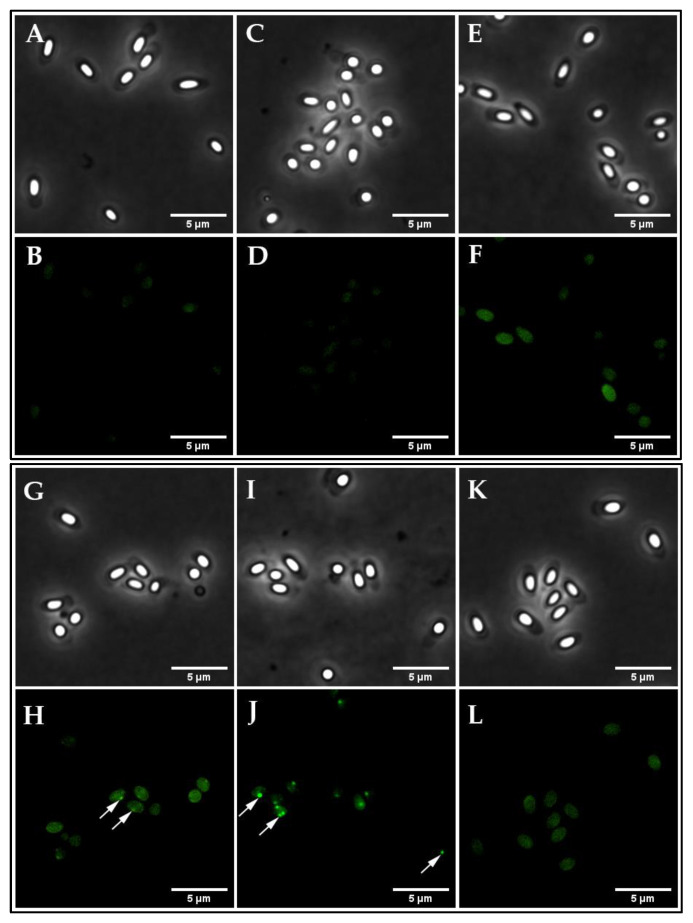
Visualization of the GerRB-SGFP2 germinosome in *B. cereus* dormant spores. Phase-contrast (**A**,**C**,**E**,**G**,**I**,**K**) and fluorescence microscopy images (**B**,**D**,**F**,**H**,**J**,**L**) of dormant spores harboring various plasmids including: (**A**,**B**) wild-type spores; (**C**,**D**) spores carrying pHT315; (**E**,**F**) spores carrying pHT315-P*aphA3′*-*SGFP2*; (**G**,**H**) spores carrying pHT315-P*gerR*-*gerRB*-*SGFP2*; (**I**,**J**) spores carrying pHT315-P*gerR*-*gerR*-*SGFP2*; (**K**,**L**) spores carrying pHT315-*gerRB*-*SGFP2* (no *gerR* promoter). Images were captured with a PH3 channel exposure time of 200 ms and an excitation channel at 470 nm using 10% laser power with an exposure time of 2 s The white arrows indicate some of the likely germinosomes in spores, and note the large heterogeneity in their fluorescence intensity. All panels are at the same magnification, and the scale bar is 5 μm.

**Figure 3 ijms-21-05198-f003:**
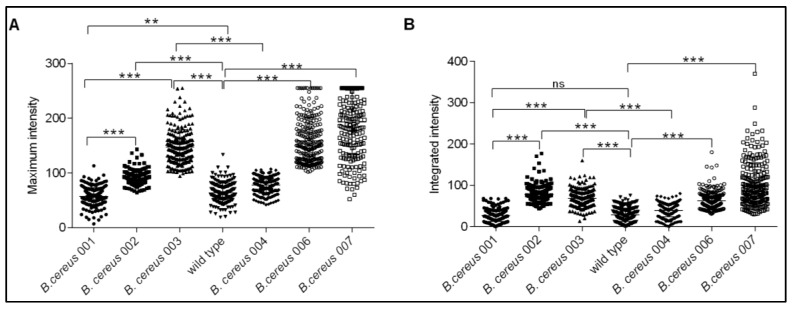
Analysis of the fluorescence intensity and calculations of fluorescence in individual *B. cereus* spores of strains numbered as in Table 1. (**A**) Maximum fluorescence intensity in dormant spores of various strains; (**B**) Integrated fluorescence intensity in dormant spores of various strains. ns: not significant, **: *p* < 0.01 and ***: *p* < 0.001. Note that in figure A the apparent dark bar at the top of the data for strain *B. cereus* 007 reflects a large number of overlapping individual points.

**Figure 4 ijms-21-05198-f004:**
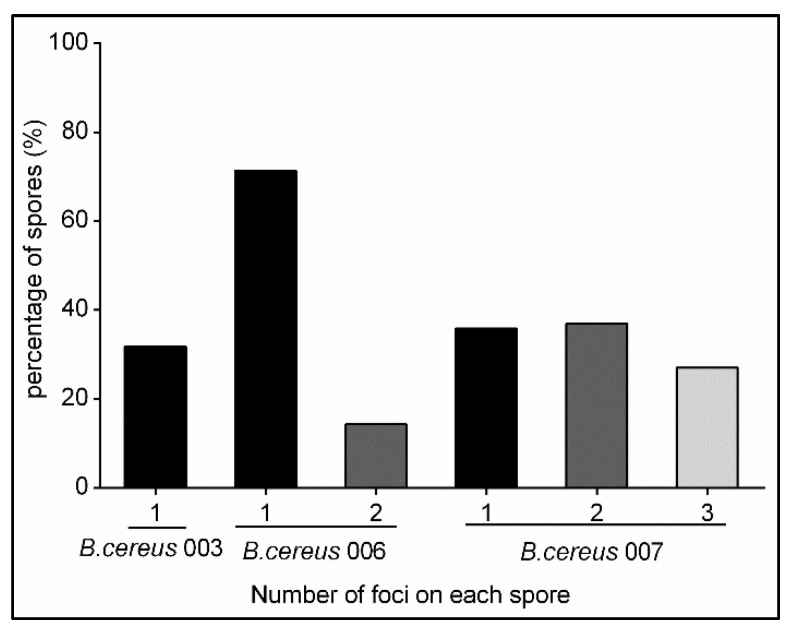
Percentage of germinosomes in dormant spores of *B. cereus* 003, *B. cereus* 006 and *B. cereus* 007. The germinosome in individual *B. cereus* 003 spores carrying pHT315-P*gerR*-*gerRB*-*SGFP2*, *B. cereus* 006 spores carrying pHT315-P*gerR*-*gerR*-*SGFP2* and *B. cereus* 007 spores carrying pHT315-P*gerD*-*gerD*-*mScarlet-I* were distinguished from the background fluorescence during the calculation process.

**Figure 5 ijms-21-05198-f005:**
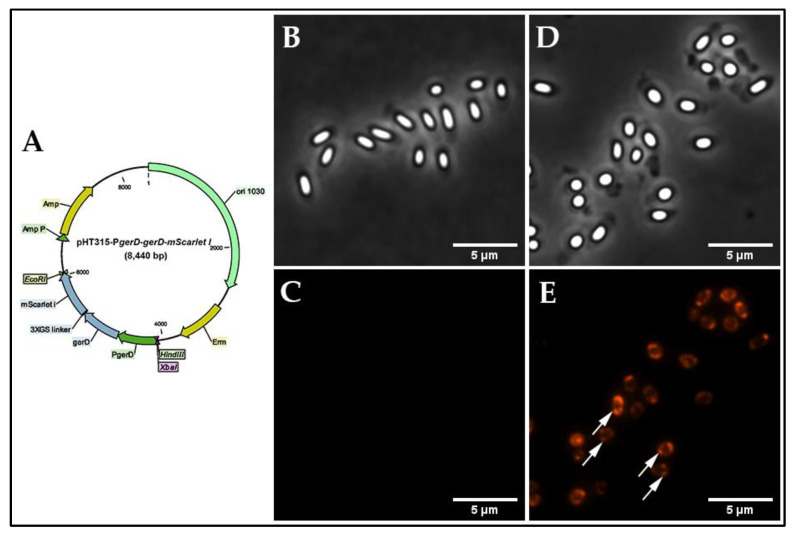
Construction of B. cereus 007 and visualization of GerD-mScarlet-I in dormant spores. (**A**), schematic diagram of plasmid pHT315-PgerD-gerD-mScarlet-I. phase-contrast (**B**,**D**) and fluorescence microscopy images (**C**,**E**) of B. cereus dormant spores: (**B**,**C**) wild-type spores and (**D**,**E**) spores carrying pHT315-PgerD-gerD-mScarlet-I. The white arrows indicate GerD in germinosomes. All panels are at the same magnification, and the scale bar is 5 μm.

**Figure 6 ijms-21-05198-f006:**
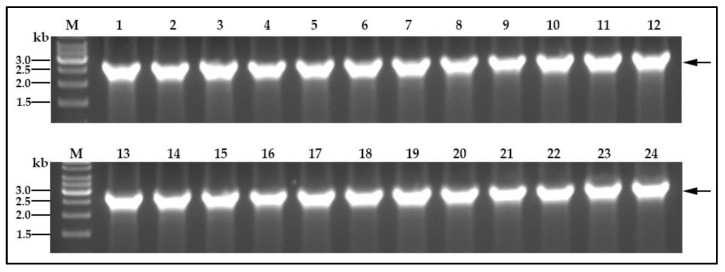
Confirmation that all germinated spores carry pHT315-P*gerR*-*gerRB*-*SGFP2*. Colony PCR was carried out on 24 independent colonies from individual spores of *B. cereus* 003 carrying pHT315-P*gerR*-*gerRB*-*SGFP2*, as described in Methods. Lanes are: M, GeneRuler 1 kb DNA ladder (Thermo Fisher Scientific); 1–24, PCR from *B. cereus* 003 carrying pHT315-P*gerR*-*gerRB*-*SGFP2*. The black arrow indicates the DNA fragment of P*gerR*- *gerRB*-*SGFP2*.

**Figure 7 ijms-21-05198-f007:**
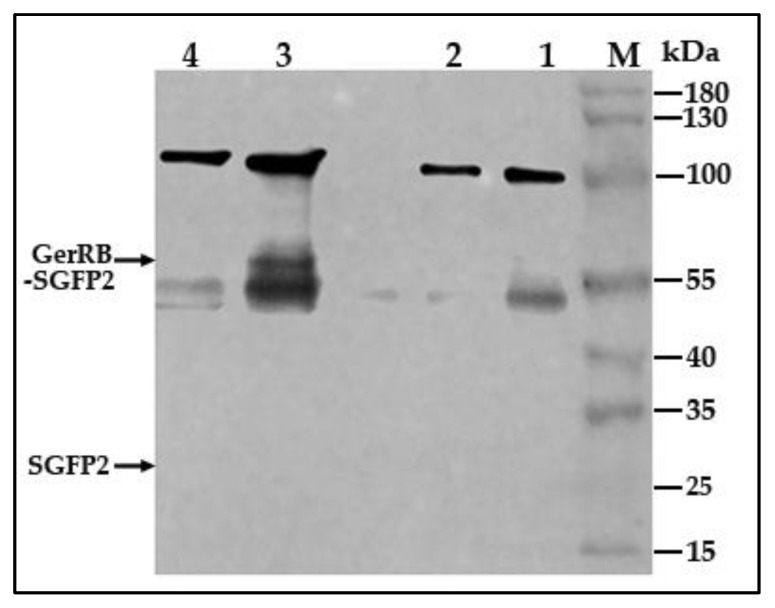
Analysis of GerRB-SGFP2 fusion protein expression in *B. cereus* spores of various strains. thirty micrograms protein in extracts from dormant spores of various *B. cereus* strains were separated on a 10% SDS-PAGE gel and western blots were probed with rabbit polyclonal anti-GFP antibody as described in Methods. The black arrow indicates the position of the expected 69 kDa GerRB-SGFP2 protein. Two nonspecific bands of ~100 kDa and ~55 kDa were also detected in all extracts. Notably, free SGFP2 at 27 kDa was not detected, in particular in extracts of spores carrying pHT315- P*gerR*-*gerRB*-*SGFP2*. The extracts run in the various lanes were from: lane 1, wild-type spores; lane 2, spores carrying pHT315; lane 3, spores carrying pHT315-P*gerR*-*gerRB*-*SGFP2*; lane 4, spores carrying pHT315-*gerRB*-*SGFP2*; lane M, prestained protein ladder.

**Table 1 ijms-21-05198-t001:** Strains used in this study.

Strains.	Genotype	Sources
*E. coli* DH5α	*F- endA1 hsdR (rk- mk+) supE44 thi-1 recA1 gyrA96 relA1*	lab stock
*B. cereus*ATCC14579	*B. cereus* wild-type	lab stock
*B. cereus* 001	*B. cereus* carrying pHT315 Ery^r^	this study
*B. cereus* 002	*B. cereus* carrying pHT315-P*aphA3′*-*SGFP2* Ery^r^	this study
*B. cereus* 003	*B. cereus* carrying pHT315-P*gerR*-*gerRB*-*SGFP2* Ery^r^	this study
*B. cereus* 004	*B. cereus* carrying pHT315-*gerRB*-*SGFP2* Ery^r^	this study
*B. cereus* 005	*B. cereus* carrying pHT315-P*gerR*-*SGFP2* Ery^r^	this study
*B. cereus* 006	*B. cereus* carrying pHT315-P*gerR*-*gerR*-*SGFP2* Ery^r^	this study
*B. cereus* 007	*B. cereus* carrying pHT315-P*gerD*-*gerD*-*mScarlet-I* Ery^r^	this study

**Table 2 ijms-21-05198-t002:** Plasmids used in this study.

Plasmids	Genotype or Description	Sources
pSGFP2-C1	source of the *SGFP2* gene Kan ^r^	lab stock
pEB2-mScarlet-I	source of the mScarlet-I gene Kan ^r^	lab stock
pHT315	a *B. thuringiensis*/*E. coli* shuttle vector Amp ^r^ Ery ^r^	[35]
pHT315-P*aphA3′*-*gfp*	expression of *gfpmut1* controlled by the constitutive P*aphA3′*promoter Amp ^r^ Ery ^r^	[35]
pHT315-P*gerR*-*SGFP2*	expression of SGFP2 was controlled by the promoter of the *gerR* operon Amp ^r^ Ery ^r^	this study
pHT315-P*aphA3′*-*SGFP2*	expression of SGFP2 was controlled by the constitutive P*aphA3′*promoter Amp ^r^ Ery ^r^	this study
pHT315-P*gerR*-*gerRB*-*SGFP2*	expression of GerRB-SGFP2 was controlled by the promoter of the *gerR* operon Amp ^r^ Ery ^r^	this study
pHT315-*gerRB*-*SGFP2*	no promoter - a negative control Amp ^r^ Ery ^r^	this study
pHT315- P*gerR*-*gerR*-*SGFP2*	expression of GerRB-SGFP2 was controlled by the promoter of the *gerR* operon Amp ^r^ Ery ^r^	this study
pHT315-P*gerD*-*gerD*-*mScarlet-I*	expression of GerD-mScarlet-I was controlled by the *gerD* promoter Amp ^r^ Ery ^r^	this study

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
