# Peer review of "Visualization of Germination Proteins in Putative Bacillus cereus Germinosomes"

_ijms, 2020, doi:10.3390/ijms21155198_

Round 1

Reviewer 1 Report

In this manuscript, the authors address an important study on Bacillus cereus germinosomes. The authors constructed the recombinant plasmids and expressed on Bacillus cereus to visualize of germination proteins. An interesting and useful paper to fine the existence of germinosomes in B. cereus spores. The aim of this manuscript is clear and the manuscript is interesting to me. I don’t have negative comments and I think it could be accepted and published on IMJS after polish English.

Author Response

Reviewer 1

In this manuscript, the authors address an important study on Bacillus cereus germinosomes. The authors constructed the recombinant plasmids and expressed on Bacillus cereus to visualize of germination proteins. An interesting and useful paper to fine the existence of germinosomes inB. cereus spores. The aim of this manuscript is clear and the manuscript is interesting to me. I don’t have negative comments and I think it could be accepted and published on IMJS after polish English.

Our reply: We thank reviewer 1 for his very positive judgement regarding our paper and went again with a fine toothcomb through the text to ‘polish the English’ some more.

Reviewer 2 Report

The manuscript presents novel data that have been carefully analysed, and demonstrate likely germinosomes in B. cereus spores for the first time. I have a small number of specific comments.

line 27. Can it be distinguished whether it improves stability, or improves localisation to the germinosome? Is there less general diffuse fluorescence if the entire operon is present?

line 35. Replace spore with endospore. Other bacterial sporeformers include myxobacteria, streptomyces etc.

line 46. Phagocytosis - from the Greek for "devouring" - this is strictly not relevant, so I don't really like to see the phrase used- engulfment is more accurate, and is already present in the sentence. I suggest removal of the two words " through phagocytosis".

line 49 layer - this word is repeated.

line 53. The core is the cellular compartment - that is not made clear.

lines 125-126 The gerD promoter is likely more active than is that of gerR, by analogy with B. subtilis, To my eye, the areas of higher fluorescence in the spores also look rather different, often being more of a wider area rather than a point focus, and often circling the spore. The introduction should also mention that GerD is likely a lipoprotein, anchored at the outer surface of the inner membrane.

line 170 Figure 2. L (no gerR promoter) seems almost as bright as F -the gfp from aph3 promoter - why? This does not seem to match the scores in Figure 3, though I accept that they show a lot of variation. 

Figure 3.  The meaning of the stars in Figure 3 should also be made explicit. The dark bar at the top of the score for the GerD fusion should be explained ( I presume it reflects a large number of overlapping individual points) 

Figure 5. Is this PCR Figure showing all spores have retained the gfp fusion really necessary in main text? It could be a supplementary figure.

Author Response

Reviewer 2

The manuscript presents novel data that have been carefully analysed, and demonstrate likely germinosomes in B. cereus spores for the first time. I have a small number of specific comments.

line 27. Can it be distinguished whether it improves stability, or improves localisation to the germinosome? Is there less general diffuse fluorescence if the entire operon is present?

Our reply: co-expression of GerRB-SGFP2 with GerRA and GerRC improves their stability. The latter is seen by us as the sum of germinosome localization and germinant receptor protein turn-over. The impression is indeed that there is less general diffuse fluorescence seen if the entire operon is present (compare figure 2H and 2J). This is now mentioned in the text of the manuscript on line 127.

line 35. Replace spore with endospore. Other bacterial sporeformers include myxobacteria, streptomyces etc.

Our reply: We thank the reviewer for this comment and have adapted the text accordingly.

line 46. Phagocytosis - from the Greek for "devouring" - this is strictly not relevant, so I don't really like to see the phrase used- engulfment is more accurate, and is already present in the sentence. I suggest removal of the two words " through phagocytosis".

Our reply: We agree with the reviewer and have removed the two words.

line 49 layer - this word is repeated.

Our reply: removed one.

line 53. The core is the cellular compartment - that is not made clear.

Our reply: We thank the reviewer for his comments and have simplified the text by omitting “is in the center, and”

lines 125-126 The gerD promoter is likely more active than is that of gerR, by analogy with B. subtilis, To my eye, the areas of higher fluorescence in the spores also look rather different, often being more of a wider area rather than a point focus, and often circling the spore. The introduction should also mention that GerD is likely a lipoprotein, anchored at the outer surface of the inner membrane.

Our reply: On lines 72, 119-120 and 130-131, we have added statements that cover these suggestions of the reviewer.

line 170 Figure 2. L (no gerR promoter) seems almost as bright as F -the gfp from aph3 promoter - why? This does not seem to match the scores in Figure 3, though I accept that they show a lot of variation. 

Our reply: Figure 2L (B. cereus strain 004 i.e. no gerR promoter) and figure 2F (B. cereus strain 002 i.e. aph3 promoter) do show differences in intensity to our eye which is reflected in the slightly elevated fluorescence trend of the latter. Importantly, the aph3 promoter is for vegetative cells a constitutive promoter of the aminoglycoside phosphotransferase gene. The promoter is not specifically activated during the sporulation process as is the gerR promoter itself. The variation in expression may at least partially be due to variable plasmid copy number in the sporulating cells.

 Figure 3.  The meaning of the stars in Figure 3 should also be made explicit. The dark bar at the top of the score for the GerD fusion should be explained (I presume it reflects a large number of overlapping individual points).

Our reply: We took notice of these comments by the reviewer and expanded the legend of Figure 3 accordingly, which dealt with these issues. 

Figure 5. Is this PCR Figure showing all spores have retained the gfp fusion really necessary in main text? It could be a supplementary figure.

Our reply: We thank the reviewer for his suggestion but prefer to keep Figure 5 in the main text, as it is a crucial piece of evidence that he/she needs to see separately. Moreover, the other two reviewers did not suggest this option.

Reviewer 3 Report

The manuscript is very useful and of interest to those involved in research on sporulation. However, the only comment I have to make is on the use of the English language. My suspicion is that various authors wrote different parts of the manuscript. Parts of the manuscript are written as if the person is speaking, rather than using proper written scientific English expression. This has to be addressed. Overall, the introduction is quite well written, there are a couple of quirks, but nothing to overly worry about.

Below I am commenting on a series of wordings that stood out. The examples are by no means an extensive editing of the text, however, someone has to go through this manuscript with a fine tooth comb and fix the English language oddities.

line 11: don't use the word 'huge', it is non-scientific

line 12: I doubt it very much that the 'food industries' aim to increase our knowledge. The food industry utilises the knowledge scientists uncover, hence it is the scientists that aim to increase knowledge

line 19: there is no need to start the sentence with 'Therefore,'. Delete the word and the sentence still makes sense.

line 21: instead of writing 'Our results were ..', write 'Our results showed ...'

line 103-104: there is no need to use the word 'end' - it is a terminal, meaning the end. Use an apostrophe for the term "spore's"

line 109 and 115: The use of the term "The hope was ..." is not appropriate scientific English. It implies that you had no idea what you were doing and gambled on an experiment. I am sure that this is not what you did.

line 114: replace the word 'whole' with 'entire'

line 117: the term 'would be more likely ...' is an odd expression. Find a more elegant way to express the meaning of the sentence.

line 120-121: Rewrite as something like "To examine the expression of the fusion proteins in spores, freshly sporulated B. cereus spores were purified and analysed by ...". It is less verbose and therefore reads easier.

Figure 4. I don't get this figure. It is mentioned in passing, but it is never properly explained. 

line 143: change 'plasmid' to 'plasmids'

lines 144-145: this sentence is very verbose

line 220: add the word 'the' between in and spore's

line 264: start the last paragraph with 'In conclusion we found ..."

line 267: start the last sentence with 'Our findings ...'

line 274: replace the word 'present' with either 'used' or 'employed'

Author Response

Reviewer 3

The manuscript is very useful and of interest to those involved in research on sporulation. However, the only comment I have to make is on the use of the English language. My suspicion is that various authors wrote different parts of the manuscript. Parts of the manuscript are written as if the person is speaking, rather than using proper written scientific English expression. This has to be addressed. Overall, the introduction is quite well written, there are a couple of quirks, but nothing to overly worry about.

Below I am commenting on a series of wordings that stood out. The examples are by no means an extensive editing of the text, however, someone has to go through this manuscript with a fine tooth comb and fix the English language oddities.

Our reply: We thank the reviewer for this suggestion and went through the text with the suggested rigour. The list of authors has extremely experienced scientists and native English-speaking members thus safeguarding the level of our scientific English used.

line 11: don't use the word 'huge', it is non-scientific

Our reply: We have replaced the word ‘huge’ with ‘serious’.

line 12: I doubt it very much that the 'food industries' aim to increase our knowledge. The food industry utilises the knowledge scientists uncover, hence it is the scientists that aim to increase knowledge

Our reply: We omitted the words ‘food industries’.

line 19: there is no need to start the sentence with 'Therefore,'. Delete the word and the sentence still makes sense.

Our reply: We thank the reviewer for his observations and omitted these words.  

line 21: instead of writing 'Our results were ..', write 'Our results showed ...'

Our reply: Done

line 103-104: there is no need to use the word 'end' - it is a terminal, meaning the end. Use an apostrophe for the term "spore's"

Our reply: Done.

line 109 and 115: The use of the term "The hope was ..." is not appropriate scientific English. It implies that you had no idea what you were doing and gambled on an experiment. I am sure that this is not what you did.

Our reply: Done.

line 114: replace the word 'whole' with 'entire'

Our reply: Done.

line 117: the term 'would be more likely ...' is an odd expression. Find a more elegant way to express the meaning of the sentence.

Our reply: We omitted would be more.

line 120-121: Rewrite as something like "To examine the expression of the fusion proteins in spores, freshly sporulated B. cereus spores were purified and analysed by ...". It is less verbose and therefore reads easier.

Our reply: We have rewritten the sentence

Figure 4. I don't get this figure. It is mentioned in passing, but it is never properly explained. 

Our reply: We now cite Figure 4 on lines 135-136.

line 143: change 'plasmid' to 'plasmids'

Our reply: Done.

lines 144-145: this sentence is very verbose

Our reply: We have simplified the sentence.

line 220: add the word 'the' between in and spore's

Our reply: Done. (now line 229)

line 264: start the last paragraph with 'In conclusion we found ..."

Our reply: Done.

line 267: start the last sentence with 'Our findings ...'

Our reply: Done.

line 274: replace the word 'present' with either 'used' or 'employed'

Our reply: Done